# Staggered Herringbone Microfluid Device for the Manufacturing of Chitosan/TPP Nanoparticles: Systematic Optimization and Preliminary Biological Evaluation

**DOI:** 10.3390/ijms20246212

**Published:** 2019-12-09

**Authors:** Enrica Chiesa, Antonietta Greco, Federica Riva, Elena Maria Tosca, Rossella Dorati, Silvia Pisani, Tiziana Modena, Bice Conti, Ida Genta

**Affiliations:** 1Department of Drug Science, University of Pavia, V.le Taramelli 12-27100 Pavia, Italy; enrica.chiesa@unipv.it (E.C.); antonietta.greco@iusspavia.it (A.G.); rossella.dorati@unipv.it (R.D.); tiziana.modena@unipv.it (T.M.); bice.conti@unipv.it (B.C.); 2Department of Public Health, Experimental and Forensic Medicine, Histology and Embryology Unit, University of Pavia, Via Forlanini 10-27100 Pavia, Italy; federica.riva01@unipv.it; 3Dipartimento di Ingegneria Industriale e dell’Informazione, University of Pavia, 27100 Pavia, Italy; elenamaria.tosca01@universitadipavia.it; 4Polymerix srl, V.le Taramelli 24-27100 Pavia, Italy; 5Immunology and Transplantation Lab, Pedriatric Hematology Oncology Unit, Department of Maternal and Children’s Health, Fondazione IRCCS Policlinico S. Matteo, 27100 Pavia, Italy; silvia.pisani01@universitadipavia.it

**Keywords:** chitosan nanoparticles, sodium tripolyphosphate, ionic gelation mechanism, microfluidics, staggered herringbone micromixer, curcumin

## Abstract

Chitosan nanoparticles (CS NPs) showed promising results in drug, vaccine and gene delivery for the treatment of various diseases. The considerable attention towards CS was owning to its outstanding biological properties, however, the main challenge in the application of CS NPs was faced during their size-controlled synthesis. Herein, ionic gelation reaction between CS and sodium tripolyphosphate (TPP), a widely used and safe CS cross-linker for biomedical application, was exploited by a microfluidic approach based on a staggered herringbone micromixer (SHM) for the synthesis of TPP cross-linked CS NPs (CS/TPP NPs). Screening design of experiments was applied to systematically evaluate the main process and formulative factors affecting CS/TPP NPs physical properties (mean size and size distribution). Effectiveness of the SHM-assisted manufacturing process was confirmed by the preliminary evaluation of the biological performance of the optimized CS/TPP NPs that were internalized in the cytosol of human mesenchymal stem cells through clathrin-mediated mechanism. Curcumin, selected as a challenging model drug, was successfully loaded into CS/TPP NPs (EE% > 70%) and slowly released up to 48 h via the diffusion mechanism. Finally, the comparison with the conventional bulk mixing method corroborated the efficacy of the microfluidics-assisted method due to the precise control of mixing at microscales.

## 1. Introduction

The great potential of polymeric nanoparticles (NPs) in treatment, prevention and diagnosis of diseases is exemplified by several decades of research and an ever-growing number of related publications in this area; nevertheless, few NPs-based formulations have achieved successful clinical translation and even fewer have received marketing approval from health authorities [1,2,3,4]. To overcome the gap and accelerate the translation from bench to bedside, unmet technical issues and concurrent regulatory aspects still need to be faced [5]. From a technical perspective, promising polymeric NPs-based formulations present unique chemistry, manufacturing and control challenges relying essentially to the choice of suitable polymeric material as well as the development of robust and easy-scalable preparative procedure in order to obtain NPs with reproducible morphological and physicochemical attributes (size, polydispersity, composition, charge and shape) greatly affecting NPs in vivo biodistribution [5,6].

A variety of biocompatible, biodegradable, synthetic or natural polymers, as such or chemically modified for tuning NPs features in terms of surface charge, drug cargo and release, active targeting or stimuli-responsive behavior have been extensively studied [7,8,9,10,11].

Among natural polymers, chitosan (CS), a linear polysaccharide composed of randomly distributed β-(1-4)-linked D-glucosamine and N-acetyl-D-glucosamine, has currently gained great attention for different applications in the biomedical field and particularly to design drug delivery systems [12]. CS is a renewable and sustainable cationic biopolymer derived from alkaline partial deacetylation of chitin, one of the most abundant biopolymers in nature and acquired from crustacean and insect exoskeletons. Additionally, CS can be obtained from enzymatic extraction of *Aspergillus niger* mycelia originated from the industrial production of citric acid and from cultivated edible mushrooms, like *Agaricus bisporus* [13]. CS is a non-toxic, biocompatible and biodegradable hydrophilic polymer with low immunogenicity that received FDA approval for wound dressings as well as in dietary application. In addition, the CS inherent antibacterial, antioxidant and mucoadhesive properties together with its capacity to transiently open epithelial tight junctions further enhance polymer versatility, making CS particularly suited for a variety of applications that include immunization and protein delivery [14], transmucosal and topical drug delivery [11,15], anti-cancer drug delivery [16], brain drug delivery [17] and gene delivery [18].

The conventional preparation techniques for CS NPs cover bulk methods based on emulsification and chemical cross-linking, emulsion droplet coalescence, emulsion solvent diffusion, reverse micellization, desolvation, ionic gelation and polyelectrolyte complexation or a combination of both [12,19]. All bulk techniques for CS NPs synthesis consist of bottom-up approaches based on the controlled assembly of molecular components forming a system with a complex structural design [4]. They provide for the use of either toxic organic or lipophilic phases, alternatively, with or without an emulsifying agent, and mostly exploit the chemically or ionically crosslinking between the CS and cross-linker molecules. Among the different CS NPs synthesis approaches, the ionic/ionotropic gelation methods are the most commonly used to obtain CS NPs because of their mild operative conditions and use of non-toxic solvents/excipients and safe cross-linkers. They involve an ionic interaction between the positively charged amino groups of CS and anionic small molecules (i.e., sodium sulphate, sodium tripolyphosphate (TPP), etc.) or polyelectrolytes (i.e., hyaluronic acid, alginates, chondroitin sulphate, sodium cellulose sulphate, etc.) and able to induce CS gelation leading to CS-based NPs formation by dropwise addition of the cross-linker solution to the CS solution or vice versa, under mild stirring and at room temperature [20,21,22,23].

Despite numerous efforts directed at process optimization, all conventional bulk techniques bring along low production efficiency, poor batch homogeneity and batch-to-batch reproducibility; furthermore, they are batchwise, time-consuming processes and the lack of automation hinders their scale-up feasibility and so clinical translation [4,5].

To overcome these issues, several passive continuous flow microfluidics-based approaches have been developed for the NPs synthesis [3,24,25]. Microfluidics—the science and technology of manipulating nanoliter volumes in microscale fluidic channels—has been commonly applied to enhance the rapid and thorough mixing of different fluids (immiscible, partially or mutually soluble), increase mass transfer across phase boundaries, and precisely control physical processes at microscales. If fluids contain NPs precursors, the three stages of NPs formation (nucleation, growth through aggregation and stabilization) are well-controlled within microchannels where aggregation can be reduced due to the continuous flow nature of the process [3,25].

In passive microfluidics techniques for NPs synthesis the proper mixing can be achieved employing special designs of the microfluidic networks and NPs physicochemical features can be tuned by simply adjusting flow rates of the fluids and therefore the mixing time, and formulation parameters such as both fluids characteristics (nature, polarity, density, viscosity, etc.) and composition (NPs precursors types and concentrations) [3,24,25]. Specifically, microfluidics has shown to be an efficient, fully-automated bottom-up technique for the synthesis of NPs with homogeneous and consistent characteristics in terms of size, size distribution, composition and drug release in a well-controlled, reproducible and high-throughput manner; low reagent consumption and short reaction time could further enable screening and optimization of libraries of NPs with customized properties [1]. Furthermore, parallelization of microfluidics platforms could enhance NPs production rate with minimal effort in newly preparation method set-up so enabling NPs formulations translation from academic research to clinical and industrial practice [1,24,26].

As far as CS-based particles synthesis is concerned, two microfluidics approaches have been investigated and mainly attributable to the microdroplet-based technology (MDT) and hydrodynamic flow focusing (HFF) [3,27]. MDT is based on the formation of an emulsion between the CS solution and an oil phase within a T-shaped microfluidic microchip; dropping the microdroplets into NaOH solution led to magnetic Fe_3_O_4_-CS microparticles with tailored shape and Ag-CS composite microparticles [28,29]. HFF is one of the most common microchip designs in microfluidics. CS-based NPs were produced employing T-shaped microchips in which the core stream of CS solution is mixed along the two adjacent sheath streams (basic water) flowing at higher flow rate. In this case hydrophobically modified-CS NPs formation was induced by polymer-controlled precipitation by the pH change [26,30,31]. In literature, this microchip design has been also employed for preparation of CS-adenosine triphosphate (CS/ATP) NPs [32,33]: in this case ATP solution (pH > 7) was introduced inside the microchip through the two adjacent streams and CS/ATP NPs were formed through ionic gelation between CS molecules and ATP, as crosslinker. 

To date, TPP can be considered the most exploitable, non-toxic and safe cross-linking agent for CS, but only one study has been reported about microfluidic-assisted TPP cross-linked CS-based (CS/TPP) NPs synthesis [34]: a complex microfluidic device was proposed consisting of four S-shape microchannel configuration in parallel with 4 inlet ports for acetic acid based CS solution, single inlet port for aqueous TPP solution and two outlet ports for the solution with synthetized NPs. Microchannels with micro-obstacle structures have been added to the microchip design in order to improve mixing efficacy.

Owing to notable potential of CS-based NPs and the poor clinical outcomes due to the troublesome preparation technique, the aim of this study is the set-up of a new microfluidic-assisted manufacturing method for the preparation of CS/TPP NPs. The microfluidic device is based on a herringbone geometry that consists of repeated pattern of grooves on the bottom of the microchannel [35]. This kind of geometry is liable of a rapid chaotic mixing inducing the stretching and folding of the fluid in laminar flow environment. Under these flow conditions the flow mixing is dominated by passive molecular diffusion and its driving force is the polymer concentration gradient. The first objective of the present study is to demonstrate, by using a systematic approach (design of experiment, DoE), the full potential of the staggered herringbone mixer (SHM) platform for promoting the synthesis of well-defined CS/TPP NPs, based on the electrostatic complexation between CS and TPP. A special focus of this work is to understand the effects of the manufacturing conditions on the CS chains assembly and thus the CS/TPP NPs compactness. To evaluate the efficacy of the new preparation method, preliminary biological experiments were carried out to determine the CS/TPP NPs biocompatibility and their cellular uptake features.

In addition, the demanding feasibility to load a hydrophobic drug into CS/TPP NPs was envisaged under the set-up microfluidic-assisted NPs preparation technique in order to further overcome the limitations of conventional bulk mixing methods. Curcumin (CURC), a poor water-soluble drug with recognized anti-inflammatory, anti-infective, anticancer, immunomodulatory, antioxidant and wound healing activities but limited pharmaceutical significance due to its chemical instability, poor bioavailability and short half-life, was selected as a model drug and challenging payload for the hydrophilic cationic CS/TPP NPs [36,37].

CURC encapsulation efficiency into CS/TPP NPs and its in vitro release profile were determined and compared to those obtained for CURC loaded CS/TPP NPs prepared by the conventional bulk mixing method.

## 2. Results

### 2.1. Synthesis of Chitosan-Based Nanoparticles by SHM Device: Preliminary Experiments

The SHM architecture was designed to create a chaotic advection mixing profile of the laminar inlet streams into the Y shape microfluidic channel. The generation of repeated rapid folding of the two streams induces an increase of the surface area at the solutions interface, an increase of the rate of mixing and then a reduction of the diffusion distance. All these factors play a crucial role in the effectiveness of system for the preparation of NPs [3,34].

The synthesis of CS/TPP NPs through the ionic gelation method involves an extremely fast mixing of two miscible aqueous solutions (acidic CS solution and basic TPP solution) into the microfluidic channel. As is already well-known, the synthesis of NPs is highly dependent on the ionic interactions involved between CS and TPP, in which their ratios can have a key role in the size of them. Moreover, CS in the presence of TPP undergoes a liquid–gel transition, increasing the compactness of the NPs modulating the drug release through the polymer matrix [38]. However, the synthesis of CS/TPP NPs by the microfluidic SHM-assisted ionic gelation method has not been systematically investigated yet.

First, preliminary experiments were carried out to define the suitable conditions for NPs formation, establishing the best concentration range of each compound to obtain a desired final product. In such concentration ranges, the NPs preparative procedure was set-up. To identify the design space in which NPs formation was observed, CS concentration were varied from 2 to 10 mg/mL, whereas CS/TPP mass ratio were ranged from 1.66/1 to 25/1. pH values of CS and TPP solutions were kept constant at 4 and 9, respectively.

Figure 1 displays the variation of the particles size, evaluated by DLS (dynamic light scattering) analysis, as a function of the CS/TPP mass ratio and CS solution concentration. As it can be noted, when the mass ratio of CS/TPP was high (20/1; 25/1 *w*/*w*), the reaction solution was clear indicating that the amount of TPP was not enough to form the cross-linked NPs. By decreasing the mass ratio (from 8.83/1 to 5/1), opalescent suspensions of particles were obtained for CS concentration lower than 5 mg/mL; particles with mean diameter below 1 µm were revealed using CS concentration lower than 2.5 mg/mL.

A further decrease of the CS/TPP mass ratio (1.66/1 and 2/1) caused aggregation phenomena leading to the formation of large aggregates. That means the mass ratio between CS and TPP is inversely related to the cross-link density of the material; a high CS/TPP ratio corresponds to a low cross-linking density providing a softer material, but also possibly a slower kinetic of formation that may be related to a more controlled process of NPs formation. From the preliminary data, the CS/TPP mass ratio between 5:1 and 8.83:1 combined to a CS concentration ranging from 2 to 2.5 mg/mL were selected as suitable design space to critically investigate the feasibility of SHM-based microfluidic device to generate CS/TPP NPs with tunable characteristics. 

### 2.2. Effect of SHM-Assisted Ionotropic Gelation Method Parameters on CS/TPP NPs Size: Design of Experiment (DoE)

To systematically assess SHM-assisted ionic gelation method feasibility for the preparation of CS/TPP NPs, a 2^3^ randomized full factorial design was used to investigate process parameters (total flow rate, TFR) and formulation variables (CS and TPP solution concentrations) effects on NPs physical properties (size and polydispersity index, PDI). Particles size is an important parameter because it can influence the biopharmaceutical properties of a NPs formulation by playing an important role in the biodistribution and release properties. PDI is a measure of the particle size distribution within a sample. CS concentration (2–2.5 mg/mL), TPP concentration (0.3–0.5 mg/mL) and TFR (5–12 mL/min) were selected as the input variable, while size and PDI were selected as the first outcomes. DoE results, repeated in triplicate for a total of 27 runs, were used to evaluate the main effect of the selected factors. Table 1 reports all the experimental data expressed as mean ± standard deviation (SD). Within the chosen design space, it was possible to produce CS/TPP particles ranging in average hydrodynamic diameter from 40 to 400 nm with reasonably acceptable distribution (except DoE batch #1,3,5).

The average Zeta (ζ) potential was always largely positive (from +18.9 ± 0.6 to +34.6 ± 1.0).

The mathematical model equation developed by software package to explain the correlation between selected factors and size is reported below:Size (nm) = 171.583 + 21.2537 × [CS] + 40.9862 × [TPP] − 108.396 × TFR + 26.3038 × [CS] × [TPP] (1)

The R-Squared statistic indicates that the model as fitted explains 95.9% of the variability in size. The adjusted R-squared statistic, which is more suitable for comparing models with different numbers of independent variables, is 91.8%. The Durbin–Watson statistic tested the residuals to determine if there is any significant correlation based on the order in which they occur in the data file. Since the *p*-value was greater than 5%, there was no correlation between the results and the order in which data were tabled.

The mathematical equation can be graphically depicted in Figure 2A where the Pareto chart represents the impact by each factor or two-factors interaction on the response comparing the mean square with the experiment standard error. The Pareto chart contains a bar for each effect, sorted from the most significant to the least significant. The length of each bar is proportional to the standardized effect. A vertical line is drawn at the location of the 0.05 critical value for the statistical test. Any bars that extend to the right of that line indicate effects that were statistically significant at the 5% significance level. In this case, two effects had *p*-values less than 0.05, indicating that they were significantly different from zero at the 95% confidence level. The significant factors were the TPP concentration *(p*-value = 0.0301) and TFR (*p*-value = 0.0010); more in detail Figure 2B shows the main effect plot where it was recognizable that an increase of TPP concentration from 0.3 to 0.5 mg/mL caused a significant enlargement of the CS/TPP NPs from 40 ± 12 to 162 ± 28 nm for DoE batches #3 and #1, from 39 ± 12 to 69 ± 4 nm for DoE batches #5 and #4, from 245 ± 26 to 305 ± 78 nm for DoE batches #8 and #6, from 225 ± 79 to 372 ± 42 nm for DoE batches #7 and #2, respectively. Instead, by increasing the TFR, a noteworthy decrease of the NPs size can be achieved: the average NPs diameters were at least two-fold reduced because of the variation of TFR from 5 to 12 mL/min.

It is reasonable that NPs sizes can be on-chip modulated by adjusting the mixing time by reaching a theoretical equilibrium between mixing rate and polymer aggregation rate. It is already explored by Nasti et al. [39] that, when the acid solution of CS (pH = 4) is mixed with a slightly basic solution of TPP (pH = 9), a transient exposure to higher pH may induce CS aggregation generating polymeric nuclei available for NPs growth. We could speculate that, after the formation of the CS nuclei due to the pH enhancement following the mixing of the CS and TPP solutions in the first segment of the SHM device, the NPs growth occurs for deposition of chitosan through electrostatic complexation with TPP. Hence, the increase of the TFR causes a remarkable reduction of the mixing time available for the CS and TPP adsorption on the nuclei surface leading to smaller CS/TPP NPs. On other hand, the increase of the TPP concentration triggers a faster complexation phenomenon liable of the NPs size rise.

The selected factors effects on the PDI is represented in Figure 2C and they can be mathematically described in the following equation: PDI = 0.435 + 0.064 × [CS] − 0.043 × [TPP] + 0.063 × TFR(2)

The mathematical model fits for 82.5% (R^2^) of the variability in PDI and two factors were significant for the sample homogeneity: TFR (*p*-value = 0.0271) and CS concentration (*p*-value = 0.0274).

Both parameters were responsible of the PDI increase (Figure 2D); an increase of the TFR led to a random NPs growth by an enhancement of the chaotic advection into the microchannel. It was also evident the influence of the CS concentration on the particles size distribution indicating a more heterogeneous polymer adsorption on nuclei surface during the mixing process: PDI values increased more than 0.1 PDI unit by varying the CS concentration from 2 to 2.5 mg/mL (Table 1). 

As indicated above, the size of CS/TPP NPs at various mixing rates is controlled by the intricate competition between intra- and inter-molecular interactions and thus it is understandable that the CS/TPP NPs sizes are governed by the polymer contents as well as their swelling state.

To gain an insight into the effect of the microfluidic preparative procedure on the characteristic of CS/TPP NPs, it is crucial assessing the NPs compactness (C_NPs_) that indicates the local polymer concentration inside the NPs. This information can be estimated by Equation (4) (see Section 3.5.3), based on Mie theory and described by Jonassen et al. [40], after the determination of NPs sizes and the turbidity of the NPs suspension.

Results, disclosed in Figure 3A, show the correlation between the C_NPs_ and the process parameters, namely TFR, CS and TPP concentrations. CS concentration clearly affected C_NPs_, by using the lowest CS concentration (2 mg/mL) more compact NPs were obtained and it can also be noted that C_NPs_ of the NPs dropped by increasing the CS concentration to 2.5 mg/mL. Moreover, excluding DoE batches #1 and #3 (blue full bar, Figure 3A), the enhancement of the TPP concentration from 0.3 to 0.5 mg/mL triggered an increase of the NP’s C_NPs_. It is reasonable considering the electrostatic forces at the base of the CS/TPP NPs formation and the crucial role of higher TPP concentration in increasing the local polymer concentration inside the particles as already highlighted for CS/TPP NPs prepared through a bulk mixing method [41,42].

Finally, the calculated C_NPs_ values for CS/TPP NPs synthetized by using different TFR revealed an enhanced local polymer concentration for the CS/TPP NPs prepared by using TFR of 12 mL/min. It is evidence of the relationship between compactness and mixing time that was also observed by Dashtimoghadam et al. [26] and it was interpreted as a competition between intra- and inter-molecular interactions, in detail an increase of the mixing time favors the probability of intramolecular association reducing the NPs compactness.

Therefore, from the abovementioned results, it was possible to conclude that low TFR is liable of either the NPs sizes increase or NPs compactness decrease. This can be due to swelling combined to the increase of the number of aggregated polymer chains inside CS/TPP NPs. To evaluate these two phenomena, the aggregation number (N_agg_) of the polymer chains was calculated from the molecular weight of the spherical NPs by using equation (6) and (7) (see Section 3.5.4). Results concerning the N_agg_, shown in Figure 3B, suggest that at high TFR the mixing occurs faster than the time scale for the polymer aggregation leading to a lower number of CS inside the NPs. Indeed, the CS/TPP NPs growth can be attributable to the higher number of CS chains associated to form CS/TPP NPs.

### 2.3. Quantification of Chitosan Recovery

CS recovery was determined by using a colorimetric assay method previously described [22]. CS recovery is useful to understand the amount of CS that effectively reacted to form NPs and to evaluate the amount of CS/TPP NPs recovered at the end of the process. For all batches performed and reported in Table 1, CS recovery values ranged from 22.5% ± 5.7 % to 62.8% ± 0.5%. As displayed by Figure 4A, a notable reduction of the CS recovery was observed when the TFR was changed from 5 to 12 mL/min (*p*-value < 0.01 and < 0.05 for 0.5 and 0.3 mg/mL of TPP concentration, respectively). The TPP concentration also had a significant impact on this outcome, its decrease allowed to a drop of the CS recovery that was lower than 40% with TFR of 12 mL/min (Figure 4B) and this behavior was confirmed at 5 mL/min (Figure 4C). Moreover, as can be noted from Figure 4C, reducing the CS concentration up to 2 mg/mL the CS recoveries had a notably rise for both the TPP concentrations tested (*p*-value < 0.05 and *p*-value < 0.0001 respectively). The highest CS recoveries (62.8% ± 0.5% and 55.5% ± 2.0%) were reached by using 5 mL/min TFR and 0.5 mg/mL TPP concentration for both the CS concentrations tested (2–2.5 mg/mL) probably because of the highest particle size (305 ± 78 nm and 372 ± 42 nm). Nevertheless, NPs recovery could be improved by increasing centrifugation force or changing the purification methods as reported for in-line tangential flow filtration system applied to the liposome manufacture by SH microfluidic device (lipid recovery higher than 90%) **[43]**. 

For the further characterization studies Doe batch #9 were used because of the superior features: suitable particles size and size uniformity (119 ± 10 nm, PDI = 0.40 ± 0.02) and acceptable CS recovery, around 49%. DoE batch #9 was characterized by a local polymer concentration (C_NPs_) of 8.79 ± 2.14 mg/mL and a N_agg_ number of 60 ± 12. TEM image of Doe batch #9 (Figure 5A) display isolated spherical CS/TPP NPs disclosing mean diameters comparable to those obtained by DLS.

### 2.4. Biological Evaluation

Aiming to prove the feasibility of the new preparation method to obtain CS/TPP NPs with successful biological behavior, preliminary biological evaluation was carried out to check the main in vitro features of the CS/TPP NPs such as cytotoxicity and the ability to be internalized.

#### 2.4.1. Cytotoxicity Test

A cytotoxicity test was carried out by using the MTT assay; five different concentrations, 2.5, 5, 12.5, 25 and 100 µg/10,000 cells, of placebo CS/TPP NPs (DoE batch #9, Table 1) were incubated with hMSCs for 24 h. Results confirmed the well-known biocompatibility of CS and the viability percentage was always higher the 80% (ranging from 82% to 93%) for all the concentration tested (Figure 5B).

#### 2.4.2. Uptake Study

It is already known by the literature that cytotoxicity of CS/TPP NPs is mostly dependent on their ability to be internalized, which is reliant not only on size but also it is ruled by charge and NPs composition [39]. Herein the ability of CS/TPP NPs to pass the cell membrane was investigated by using RhB labeled CS/TPP NPs. RhB conjugated CS (CS-RhB) was successfully synthetized; the labeling efficiency was 0.267 µg of RhB/ µg of CS and moreover UV-vis spectrum of CS-RhB (Figure 6A) shows the characteristic peak of RhB at 554 nm confirming the effective conjugation between CS and RhB.

CS-RhB/TPP NPs were prepared through SHM microfluidic device by setting the TFR at 8.5 mL/min and FRR at 1:1 (*v*:*v*), by using a suitable blend of CS (98% wt) and CS-RhB (2% wt) to reach the final CS concentration of 2.25 mg/mL. The CS-RhB/TPP NPs size was comparable to that of the blank one and the positive surface charge was maintained (+21.3 ± 2.8 mV).

hMSCs (20,000 cells), used as model cells, were incubated with 50 µg and 100 µg of CS-RHB/TPP NPs for 30, 90 and 240 min. As shown in Figure 6B, CS/TPP NPs are quickly internalized by hMSCs: CS/TPP NPs can already pass the cell membrane at 30 min of incubation and after 240 min they are massively located into the cell cytosol. This evidence is confirmed by the quantitative evaluation of the red fluorescence inside the cells (Figure 6C). Statistical analysis revealed a noteworthy increase of the CS/TPP NPs uptake at 240 min of incubation compared to 30 min and 90 min *(p*-value < 0.0001), conversely no statistical differences were shown between the different amounts of NPs tested (50–100 µg/20,000). Figure 6D shows the 3D projects of the hMSCs culture after 240 min of incubation with NPs. 3D projects were obtained by the elaboration of the confocal images along the z-axis and, as it can be seen, the CS-RhB/TPP NPs were widely distributed into the cytosol near the perinuclear region.

Interestingly, in the 3D project of the hMSCs culture incubated with CS-RhB/TPP NPs for 30 min (Figure 7A) some cell membrane protrusions were observed as highlighted in the with square, and these entities were better investigated in Figure 7B that reveals the presence of endocytic vesicles arising from the cell membrane. More in detail, these endocytic vesicles can include CS-RhB/TPP NPs as demonstrated in Figure 7C by assessing the red and green fluorescence intensities along the length of line that cross the selected vesicle. The peak of the red fluorescence intensity is in the middle of the two peaks corresponding to the green fluorescence intensity.

Despite of the new preparation method, the CS-RhB/TPP NPs can successfully enter the cells through an unspecific electrostatic interaction with the negatively charged component of the cellular membrane, possibly utilizing clathrin-mediated mechanism where, however, interaction with membrane-linked GAGs may play a role too [44,45].

### 2.5. Environmental Effects on CS/TPP NPs

The swelling behavior of CS/TPP NPs was determined at pH 5.0 and pH 7.4. The CS/TPP NPs mean size were maintained up to 90 min of incubation then the swelling in the more evident in acidic environment pH 5.0 than at pH 7.4. The swelling index was determined at 24 h of incubation as 54% and 42% at pH 5.0 and pH 7.4, respectively. The enhanced swelling of NPs at pH 5.0 is principally due to the superior swelling behavior of CS at acidic pH, particularly below its pKa 6.2. At these pH values, the highly protonated amino groups repel each other creating cavities, thereby permitting the water moieties to enter the NPs core and swells. On the contrary, at higher pH, CS molecules shrink due to low protonation and thus the NPs sizes were slightly reduced.

### 2.6. Effect of the Encapsulation of a Hydrophobic Model Drug: Curcumin

We demonstrated that the SHM device allowed for a size-controlled and extremely fast synthesis of CS/TPP NPs; to investigate the effective applicability of this microfluidics technology platform for a high-throughput NPs production, the drug encapsulation efficiency is a crucial feature to be considered. Placebo DoE batch #9 (Table 1) was selected for the further studies on the drug encapsulation capability. Known the CS hydrophilicity, a water-insoluble small molecule (CURC) was used as model drug in order to evaluate the worst and most challenging case for drug encapsulation. Aiming to load CURC, ethanol (5% *v*/*v*) was added to CS aqueous solution and placebo batches were prepared accordingly with Table 1. After the addition of ethanol, a slight increase in particles size was detected: CS/TPP NPs had mean diameter of 164 ± 33 nm.

Instead, a superimposable positive surface charge (+19.7 ± 3.3 mV) was displayed. Two different CURC concentrations were tested, namely 9 µg/mL (batch CURC#1) and 4.5 µg/mL (batch CURC#2); the addition of the CURC did not influence the NPs sizes that was 157 ± 41 nm and 172 ± 71 nm for batches CURC#1 and CURC#2, respectively.

For both batches CURC#1 and CURC#2 drug encapsulation results were satisfactory: CURC encapsulation efficiency was 74.81% ± 2.78% and 75.99% ± 0.24%, respectively. The different CURC concentrations (9 µg/mL and 4.5 µg/mL) used for NPs batches formation did not affect the percentage of the entrapped drug.

#### Conventional Bulk Mixing Ionotropic Gelation Method

In order to highlight the advantages and tunable performances of the SHM device in the production of CS/TPP NPs, CURC loaded CS/TPP NPs were prepared by a previously set-up bulk mixing method [46]. It consists of several sonication and stirring cycles liable to produce homogeneous NPs, however this type of NPs manufacturing method was more time-consuming and complicated, further the operator variability cannot be overlooked. As reported in Table 2 by using the conventional bulk mixing method CURC loaded CS/TPP NPs with a mean size of 113 ± 15 nm were obtained with acceptable values of PDI (0.36 ± 0.05). Instead, CURC loading efficiency resulted to be 37.12% ± 10.34%, significantly reduced compared to that obtained with the SHM device.

### 2.7. Curcumin Release Studies

CURC release profile was evaluated in PBS at pH 7.4 to mimic the physiological condition; CTAB (0.5% *w*/*v*) was added to the buffer in order to enhance CURC solubility and stability in the release medium [47,48]. Drug release profiles from CS/TPP NPs, prepared by using both SHM device and bulk mixer, were compared.

The CURC release profile from CS/TPP NPs is reported in Figure 8A. Compared to the CURC dissolution profile, its release from CS/TPP NPs produced by SHM was prolonged up to 2880 min (48 h) and interestingly, a low burst release was observed: after 4 h (240 min) of incubation only 17% of the drug entrapped was released. On the other hand, CURC release profile from bulk mixing-made CS/TPP NPs was characterized by a notable burst effect: 60% of the entrapped drug was released in the first 30 min of incubation reaching the 80% at 8 h of incubation (Figure 8A). The release of the drug was completed at 24 h. To investigate the CURC release kinetic, release data derived by SHM-made CS/TPP were fitted, as (i) cumulative amount of drug released versus time (zero order kinetics model), (ii) ln cumulative percentage drug release versus time (first-order kinetics model) and (iii) cumulative percentage drug release versus square root of time (Higuchi kinetic model). The goodness-of-fit of the model was evaluated by the correlation coefficient; R^2^ of 0.98 highlighted the experimentally observed data are well linearized by Higuchi’s model (Figure 8B). This kinetic pattern indicated that release was dominated by the diffusion model, which normally depends on the drug concentration gradient between NPs and dissolution media with penetration of this media through a porous wall.

## 3. Materials and Methods

### 3.1. Materials

Chitosan chloride salt (Chitoceuticals) with a deacetylation degree of 82.2% and viscosity (1% in water) of 19 MPas was purchased by Heppe Medical Chitosan GmbH (Halle, Saale-Germany. Size exclusion chromatography revealed a weight-average molecular weight of 282,000 Da and a polydispersity index of 1.22. Sodium tripolyphosphate (TPP), Cibacron Brilliant Red 3B-A (dye content 50%), cetyl trimethylammonium bromide (CTAB), Dulbecco’s modified eagle’s medium–high glucose (DMEM), Dulbecco’s phosphate buffered saline (PBS 10X, sterile), Hoechst 33258 solution, Rhodamine B (RhB) and penicillin–streptomycin solution (100X) were obtained from Sigma Aldrich (St. Louis, MO, USA). Fetal bovine serum was obtained by EuroClone Spa (Milan, Italy). Anti-CD44 (Phagocytic glycoprotein-1, HCAM) monoclonal mouse primary antibody anti-human CD44 and anti-mouse IgG secondary antibody FITC conjugated were supplied from Biogenex (San Ramon, CA, USA) and Sigma-Aldrich (St. Louis, MO, USA), respectively.

The water used in the preparation of polymeric solutions was distilled and filtered through 0.22 μm membrane filters (Millipore Corporation, Billerica, MA, USA); all other chemicals were of analytical or HPLC grade.

#### Cell Lines

All biological tests were performed using mesenchymal stem cells from human bone marrow (hMSCs), kindly provided by the Pediatric Oncohematology Unit (Fondazione IRCCS Policlinico San Matteo, Pavia, Italy). hMSCs cells were cultured at 37 °C with 5% CO_2_ in 25 cm^2^ culture flask with DMEM containing 1% (*v*/*v*) penicillin–streptomycin and 10% (*v*/*v*) FBS.

### 3.2. Microfluidic Device

A staggered herringbone micromixer (SHM) with the automated mixing platform NanoAssemblr^™^ was purchased by Precision NanoSystems Inc. (Vancouver, Canada). The NanoAssemblr^™^ platform is composed of disposable SHM cartridges, a syringe pump and NanoAssemblr^™^ software. Cartridges with dimensions of 6.6 cm × 5.5 cm × 0.8 cm (w × d × h) was made of poly-propylene, viton and cyclic olefin copolymer and it is characterized by a Y-shaped architecture incorporating staggered herringbone ridges. Cartridge’s mixing channels were 200 μm × 79 μm (w × h) and the herringbone structure was 31 μm high and 50 μm thick. An angle of 45° divides the ridges and the long axis of the channel [49].

A mixing cycle is composed of two sequential regions of ridges with herringbone structure; the direction of asymmetry of the ridges switches with respect to the centerline of the channel from one region to the next. The ridges allow producing chaotic flow; these stirring flows will reduce the mixing length by decreasing the average distance over which diffusion must act in the transverse direction to homogenize unmixed volumes [35]. Briefly, solutions are injected into separate inlet points on the Y-shape micromixer using disposable and compatible syringes connected directly on the cartridge. Upon selecting the suitable process parameters, total flow rate intended as the sum of the flow rates of two inlet solutions (TFR) and the relative flow rate ratio (FRR) indicative of the volume ratio between the two inlet solutions, sample batches are collected from the outlet point, with the first and the last droplets recovered separately as waste. The mixing is dominated by the passive molecular diffusion of the solutes into the two streams based on their concentration gradient. Therefore, the mixing time derived from the combination of the diffusion time, depending on FRR as described from Karnik et al. [50], and the flow time inside the microchannel, expressed as TFR.

### 3.3. Synthesis of Chitosan-Based Nanoparticles by SHM Device: Preliminary Experiments

CS and TPP were dissolved separately in distilled water and the NanoAssemblr^™^ platform equipped with the SHM was used to prepare placebo CS/TPP NPs. An aliquot of 1.5 mL of CS aqueous solution ranging from 2 to 10 mg/mL was mixed into the microchannel with 1.5 mL of TPP aqueous solution at increasing concentration from 0.08 to 6 mg/mL by using disposable syringes connected to the inlets of the microfluidic cartridge. Of the sample 3 mL was recovered from the outlet, the initial and end waste were set at 0.350 and 0.050 mL respectively, indeed the core sample was 2.6 mL. Different CS/TPP weight ratios were screened from 1.66/1 to 25/1 to identify the suitable condition for the NPs formation avoiding the formation of aggregates. TFR was kept constant at 8.5 mL/min.

### 3.4. Effect of SHM-Assisted Nanoprecipitation Method Parameters on CS/TPP NPs Size: Design of Experiment (DoE)

DoE formulation screening study was performed by using Statgraphics Centurion XVII to identify how the critical process and formulation variables and their interactions influence the sizes and size distribution (PDI) of placebo CS/TPP NPs.

Full factorial design (FFD; 8, 23) was used to evaluate the significant effect of process parameters (input) on the selected outputs. For each input was defined a minimum and maximum level (–1 and +1 respectively) and a central point (0), that is the mean value of each input. CS concentration, TPP concentration and the TFR were selected as inputs and the selected screening design was performed in triplicate resulting in 27 runs organized in randomized way. CS concentration was varied from 2 to 2.5 mg/mL while selected TPP concentrations were 0.3 and 0.5 mg/mL. Lastly, TFR was set at 5 and 12 mL/min keeping constant FRR at 1:1 *v*/*v*. Mean size and size distribution were chosen as critical output.

Experimental results were used for the model validation by an ANOVA analysis indicating the goodness of the model prediction and the regression coefficient of each input established its statistical significance.

### 3.5. CS/TPP NPs Characterization

#### 3.5.1. Size, Surface Charge and Morphology

Volume-average particle diameter and polydispersity index (PDI) were measured by using dynamic light scattering (DLS) NICOMP 380 ZLS apparatus (Particle Sizing System, Menlo Park, CA, USA). After the CS/TPP batch preparation 2.5 mL of NPs suspension (about 1 mg/mL) was loaded in the detection cell and analyzed. For zeta potential analysis, all analyses were carried out by electrophoretic light scattering (NICOMP 380 ZLS apparatus, Particle Sizing System, Menlo Park, CA, USA) after samples dilution with 10 mM NaCl aqueous solution. All measurements were made almost in triplicate and results were expressed as mean values ± standard deviation (SD).

NPs morphology was evaluated by transmission electron microscopy (TEM, EM-2085, Philips, Eindhoven, Holland). A 1% wt uranyl acetate aqueous solution was used as the contrast enhancing solution. Of the NPs aqueous suspension mixed with the contrast agent 20 µL was deposited on the top of the copper grid, then the excess of reagent was removed by filter paper.

#### 3.5.2. Turbidimetry

The transmittance of CS/TPP NPs suspensions was measured by using spectrophotometry (6705 UV-vis spectrophotometer, Jenway, Staffordshire, UK) at room temperature. The turbidity (τ) was calculated from the transmittance through the Beer–Lambert law:τ = −1/L × ln(I/Io)(3)
where L, I and Io are the thickness of the cell, the intensity of the light transmitted through the sample and the intensity of the light transmitted to the solvent, respectively.

The measurements were carried out at 635 nm, as reported in Dashtimoghadam et al. [26], and analyses were repeated in triplicate.

#### 3.5.3. CS/TPP NPs compactness (CNPs)

The local polymer concentration inside the NPs, C_NPs_, was calculated exploiting the following equation:(4)$$\tau \;=\;\frac{3c_{t}}{2R_{h}c_{NPs}}\;\left[1-\;\frac{2}{wc_{NPs}}\left(\sin\left(wc_{NPs}\right)\right)-\frac{1}{wc_{NPs}}\left(1-\cos\left(wc_{NPs}\right)\right)\right]$$
where c_t_ is the total polymer concentration in suspension, R_h_ is the hydrodynamic radius and w is represented by the following equation:
w = [4πR_h_ (dn/dc)]/λn_0_(5)
where n_0_ is the refractive index of the solvent and dn/dc is the refractive index increment of the polymer, for the CS with superimposable characteristics of that used in this work, the refractive index increment value was found to be 0.157 g/mL [41]. This method is developed for spherical, monodisperse particles and therefore the CS/TPP NPs are assumed to be spherical.

#### 3.5.4. Aggregation Number (N_agg_) of CS Chains in the CS/TPP NPs

After the determination of C_NPs_, the molecular weight of the spherical nanoparticles M_NP_ can be calculated by using the following expression:(6)$$M_{\mathrm{NP}}\;=\;\frac{4}{3}\;\pi R_{h}^{3}C_{NPs}N_{A}$$
where NA is the Avogadro’s number. The aggregation number of polymer chains in the NPs (N_agg_) can be evaluated by using the equation below:(7)$$N_{\mathrm{agg}}=\;\frac{M_{NP}}{M_{n}}$$
where Mn is the number average molecular weight. Chitosan used for this study had a M_n_ of 233,346 Da (determined by gel permeation chromatography).

#### 3.5.5. Quantification of Chitosan Recovery

A Cibacron Brilliant Red 3B-A colorimetric assay method [22,51] was used to evaluate the amount of CS that reacts to form an electrolytic complex with TPP leading to NPs. After the NPs suspension centrifugation (at 25000 rcf for 20 min, Eppendorf centrifuge 5417 R, Eppendorf, Milano, Italy), the supernatant (100 µL) was diluted with 200 µL of glycine buffer and 3 mL of dye solution (1.5 g/L). The sample was analyzed by using a UV-Vis spectrophotometer (6705 UV-vis spectrophotometer, Jenway, Staffordshire, UK) at 575 nm. The reference solution consisted of a mixture made of glycine buffer (300 µL) and dye solution (3 mL). An unknown sample was determined against chitosan standard solution in a concentration range 2.27–37.87 µg/mL (slope = 0.0162, intercept = 0.0298, R^2^ = 0.9939).

CS recovery percentage was calculated as follows:%CS = 100 × (CS_(tot)_ − CS_(sup)_)/(CS_(tot)_)(8)
where CS_(tot)_ is the starting raw materials (mg) while CS_(sup)_ is the no-reacted CS amount quantified in the supernatant.

### 3.6. Biological Evaluation

#### 3.6.1. Cytotoxicity Test

Different amounts of placebo NPs were incubated with hMSCs to determine their toxicity. The effect of CS/TPP NPs was assessed by MTT assay using 96-well cell culture cluster with 10,000 cells in contact to different amounts of NPs, from 2.5 to 100 µg.

#### 3.6.2. Uptake Studies

Cellular uptake of placebo Rhodamine-conjugated (RhB) CS/TPP NPs, prepared as described in Section 3.6.2.1, was evaluated by confocal microscopy (Leica TCSSP8, AOBS, Germany). hMSCs were seeded on the bottom glass slide at the density of 20,000 cell/well and cultured (37 °C, 5% CO_2_) in DMEM with FBS (10% *v*/*v*) and antibiotics (1% *v*/*v*) until reaching cells confluence. Then, cells were treated with 500 µL of fluorescence CS/TPP NPs suspension at different concentrations namely 50 µg and 100 µg/20,000 cells; NPs and cells were incubated for 30, 90 and 240 min. At scheduled time points, all media were removed, hMSCs were first washed and 3.then fixed with 4% wt paraformaldehyde aqueous solution.

Cell nuclei were stained by Hoechst33258 (Sigma Aldrich, St Luis, MO, USA). Cell membrane was observed highlighting the CD44 expression on the hMSCs surface by immunocytochemistry assay as previously described [52]. Cells were observed by confocal microscopy (obj mag 63×) and all confocal images were elaborated by using ImageJ software to evaluate the CS/TPP NPs uptake in hMSCs.

##### Fluorescent CS-RhB/TPP NPs Preparation

RhB (490 mg, 0.001 mol) was grafted to CS (200 mg, 0.7 µmol) through an amidation reaction by using 1-Ethyl-3-[3-(dimethylamino) propyl] carbodiimide hydrochloride (EDC, 390 mg, 0.002 mol) and N-hydroxysuccinimide (NHS, 230 mg, 0.002 mol) as a reagent to form CS-RhB conjugate with red fluorescence. The conjugate was purified by dialysis (M_wco_: 12,000–14,000 Da, Spectra/Por^®^) against demineralized water for three days in the dark. Finally, the purified product was freeze-dried.

The UV-vis spectrum of the fluorescent CS-RhB conjugate solution (5 μg/mL), scanned from 200 to 700 nm, was compared to RhB solution (5 μg/mL) spectrum confirming the CS-RhB conjugation (Figure 5A). The RhB labeling efficiency was determined by evaluating the absorbance at 554 nm of the CS-RhB conjugate solution against RhB standard solutions in a rank of 0.25–4 µg/mL (R^2^ = 0.9991).

To prepare CS-RhB/TPP NPs, CS (98% wt) and CS-RhB (2% wt) were dissolved in water to reach the final concentration of 2.25 mg/mL while TPP aqueous solution concentration was 0.4 mg/mL. Sample preparation was carried out as reported in Section 3.3 exploiting the microfluidic device. TFR and FRR were set at 8.5 mL/min and 1:1, respectively.

### 3.7. Environmental Effect on CS/TPP NPs

The pH-dependent swelling behavior of the CS/TPP NPs was studied at pH 5 (acetate buffer) and pH 7.4 (PBS buffer) for 24 h. Of CS/TPP NPs suspension (1.125 mg/mL) 2.6 mL was dialyzed against 30 mL of each buffer under magnetic stirring. At a scheduled time point of 30, 90, 240 and 1440 min (24 h) NPs size was checked by DLS analysis. Results were expressed as percentage swelling (% SW) calculated as follows:%SW = 100 × (d_0_ − d_t_)/ d_0_(9)
where d_t_ is the NPs mean diameter at the incubation time point while d_0_ is the starting NPs mean diameter.

### 3.8. Synthesis of Curcumin Loaded CS/TPP NPs by SHM Device

To prepare CURC loaded CS/TPP NPs, an aliquot of CURC ethanolic solution (0.18 and 0.09 mg/mL) was added to the CS solution: the final ethanol percentage was 5% *v*/*v* with a CURC final concentration of 9 and 4.5 µg/mL. The hydroalcoholic mixture of CS and CURC was injected into the SHM device, separately from the TPP aqueous solution: FRR was kept constant at 1:1 (*v*/*v*) and TFR was set at 8.5 mL/min. NPs were recovered from the outlet and purified by centrifugation at 25,000 rcf for 20 min (Eppendorf centrifuge 5417R, Eppendorf, Milano). Supernatant was analyzed by UV-vis spectrophotometer measuring the absorbance at 425 nm to determine the amount of CURC loaded into CS/TPP NPs. Standard curve (slope = 0.1462, intercept = 0.0025, R^2^ = 0.9995) pre-constructed with serial dilution of CURC (from 0.5 to 6 µg/mL) were used for the conversion of absorbance to CURC concentration. Results were expressed as encapsulation efficiency (%EE) ± SD, averaged over at least six independent experiments.

### 3.9. Synthesis of Curcumin Loaded CS/TPP NPs by Bulk Mixing Ionotropic Gelation Method

In order to highlight actual advantages and tunable performance of SHM device, placebo and CURC loaded NPs were prepared by a set up conventional bulk mixing ionotropic gelation method as well.

Placebo CS/TPP NPs were obtained by adding 2 mL of TPP aqueous solution (0.75 mg/mL) dropwise into 10 mL of CS aqueous solution (1 mg/mL) under gentle magnetic stirring for 20 min. Then, NPs suspension underwent three sonication cycle in bath sonicator (Sonica^®^, Soltec s.r.l., Milan, Italy) of 10 min each. Each sonication cycle was interspersed with magnetic stirring for 10 min into ice bath to prevent temperature increase.

To prepare curcumin loaded CS/TPP NPs, 150 μL of CURC ethanolic solution (0.18 mg/mL) was previously added to the TPP aqueous solution to be poured into the CS solution following the aforementioned protocol.

CS/TPP NPs were then recovered by centrifugation at 16,400 rpm for 20 min. Supernatants were analyzed as describe in Section 3.8.

### 3.10. Curcumin Release Studies

The drug release profile from the CS/TPP NPs was assessed as follows: 1 mg of CURC loaded CS/TPP NPs was incubated at 37 °C under gentle shaking with 2 mL of 0.5% *w*/*v* CTAB solution (in 10 mM PBS, pH 7.4). Four samples of CURC loaded CS/TPP NPs, corresponding to a CURC amount of around 3, 5 and 10 µg, were employed for this test. At scheduled time points, the samples were centrifuged (25,000 rcf for 20 min) and then the supernatant were collected and analyzed by UV-vis spectrophotometer as previously reported (see Section 3.8) to evaluate the released drug amount.

Results were expressed as mean CURC release percentage ± SD (*n* = 3); curcumin as such underwent a dissolution test in the same experimental conditions.

### 3.11. Statistical Analysis

Data were expressed as mean ± standard deviation (SD). The statistical significance of the differences was determined of two-way analysis of variance (ANOVA). Differences were considered significant at *p*-value < 0.05. All statistical analyses were performed in GraphPad Prism version 6 (GraphPad Software Inc., La Jolla, CA, USA)

## 4. Conclusions

CS is one of the most sought-after biomaterials for the synthesis of NPs aimed to drug delivery application. Despite the countless examples stated in the literature, reproducibility is often an issue. The ionic gelation method based on the electrostatic interaction between amine groups of CS and negatively charged phosphate group of TPP employs mild conditions suitable for drug delivery involving the dropwise addition of the cross-linker (TPP) to the polymer (CS) solution by slow, uncontrolled bulk mixing. The lack of control over the mixing in the conventional bulk synthesis method typically resulted in an unreproducible synthesis method and in the poorly defined NPs that further affects their physicochemical properties as loading and controlled release. Microfluidics emerged as an effective platform for the synthesis of various NPs, but only few papers referred to CS/TPP NPs preparation.

Herein, we demonstrated the feasibility of the SHM-assisted synthesis to produce CS/TPP NPs with tailored made physical properties. First, by using a rational experimental design (DoE) we studied the influence of several factors as TFR, CS and TPP concentration on the NPs mean size and the size polydispersity; more in detail mean size was affected by the TFR and TPP concentration while the sample homogeneity was influenced by either CS concentration and TFR. Indeed, it is pivotal reaching a suitable equilibrium between the formation rate of the polymeric nuclei triggered to the exposure of CS to the basic pH of TPP solution and the electrostatic complexation rate of CS and TPP. Particles compactness, namely the local polymer concentration inside the NPs is influenced by the starting CS–TPP concentration and the TFR. More in detail high TFR (12 mL/min) lead to compacter and smaller CS/TPP NPs due to the faster mixing time that plays an important role in the NPs formation. A slow mixing rate is responsible of the CS intramolecular associations that reduce the CS/TPP compactness and the high number of kinetically locked CS inside the NPs. CS recovery was satisfactory ranging from 22.46% ± 5.72% to 62.81% ± 0.45%, however it should be improved by changing the purification process as suggested by Forbes et al. for liposomes manufactured by the SHM microfluidic device and recovered by an elegant in line purification method [43].

As a result of the application of a robust approach, CS/TPP NPs with mean size of 119 ± 10, PDI lower than 0.4 and positive surface charge (+25 mV) were selected for a deeper characterization. CS/TPP NPs are effectively internalized into the cytosol of hMSCs starting from 30 min of incubation via the endocytic pathway exploiting the unspecific electrostatic interaction between CS/TPP NPs and negative compounds of the cell’s membrane.

CURC, a poor water-soluble drug used as the worst model payload, was successfully loaded into the selected CS/TPP NPs (EE% higher than 70%). The encapsulation efficacy achieved by the SHM device was significantly higher than that obtained by the conventional bulk mixing method (%EE = 37.12% ± 10.34%). Moreover, the CURC release profile is noteworthy influenced by the preparative procedure, the burst release was dramatically reduced by using SHM device and the CURC release was completed at 48 h of incubation.

These results demonstrated that the SHM-assisted ionotropic gelation method was a robust, versatile and well reproducible technique to precisely induce electrostatic interaction between polymer and its crosslinker with tailor made size. This approach could be extended to another type of natural polymer-based NPs.

## Figures and Tables

**Figure 1 ijms-20-06212-f001:**
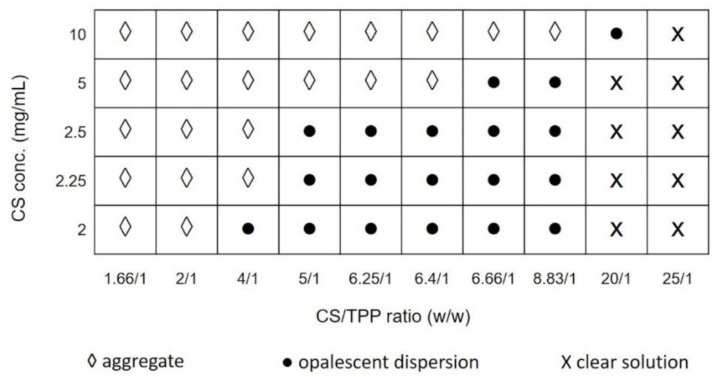
Phase diagram of nanoparticles (NPs) formation, using chitosan (CS) solution at different concentrations and for different CS/TPP mass ratios, with three areas: clear solution (X), opalescent suspension (●) and aggregates (◊).

**Figure 2 ijms-20-06212-f002:**
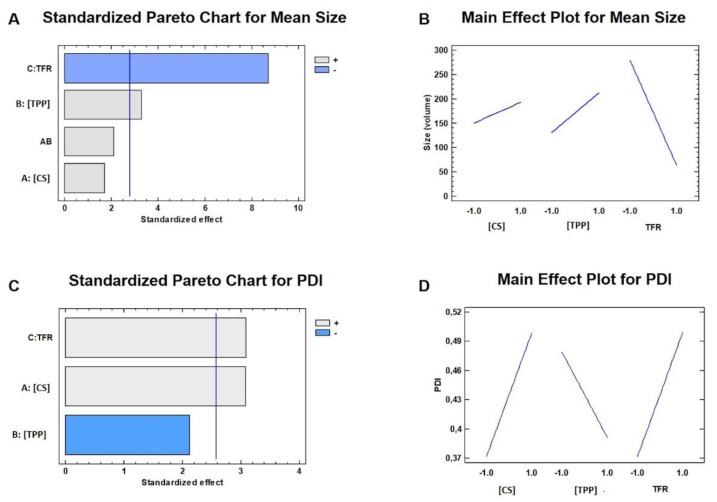
Optimization of the manufacturing process by DoE: Pareto charts and main effect plots for the mean size (**A**,**B**) and for the PDI (**C**,**D**).

**Figure 3 ijms-20-06212-f003:**
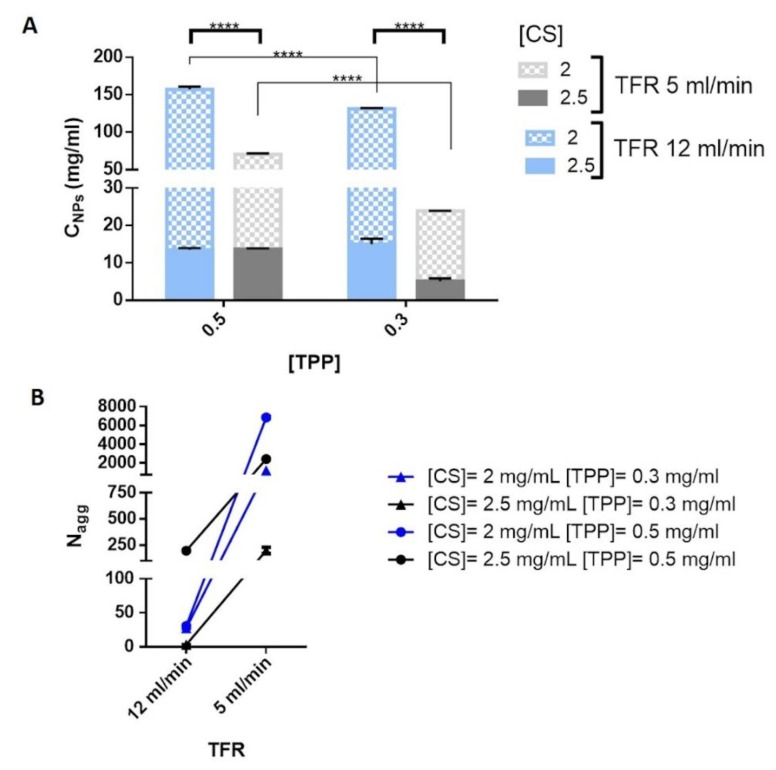
(**A**) Local polymer concentration (C_NPs_) inside CS/TPP NPs as a function of TFR, CS and TPP concentrations ([CS] and [TPP]). Sidak and Tukey’s multiple comparison tests reveal significant differences (****) for *p*-value < 0.0001. (**B**) Aggregation number (N_agg_) of CS chains in the CS/TPP NPs as a function of TFR.

**Figure 4 ijms-20-06212-f004:**
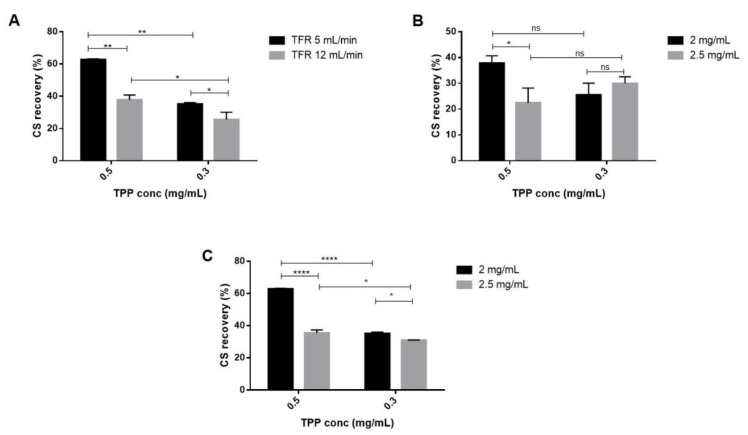
(**A**) CS recovery as a function of TPP concentration (0.3–0.5 mg/mL) and TFR (5–12 mL/min) keeping constant CS concentration (2 mg/mL). (**B**) CS recovery as a function of TPP concentration (0.3–0.5 mg/mL) and CS concentration (2–2.5 mg/mL) keeping constant TFR 12 mL/min. (**C**) CS recovery as a function of TPP concentration (0.3–0.5 mg/mL) and CS concentration (2–2.5 mg/mL) keeping constant TFR 5 mL/min. Sidak’s multiple comparisons test reveals significant differences (*) for *p*-value < 0.05, (**) for *p*-value < 0.01 and (****) for *p*-value < 0.0001.

**Figure 5 ijms-20-06212-f005:**
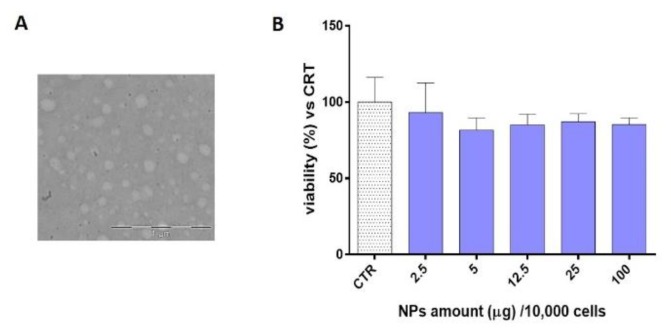
(**A**) TEM image of CS/TPP NPs (DoE Batch #9), obj. mag 30 kX. (**B**) In vitro cytotoxicity of CS/TPP NPs determined by MTT assay on hMSCs after 24 h of incubation. Results are expressed as cell viability percentage vs. CTR (untreated cells) mean ± SD (*n* = 3).

**Figure 6 ijms-20-06212-f006:**
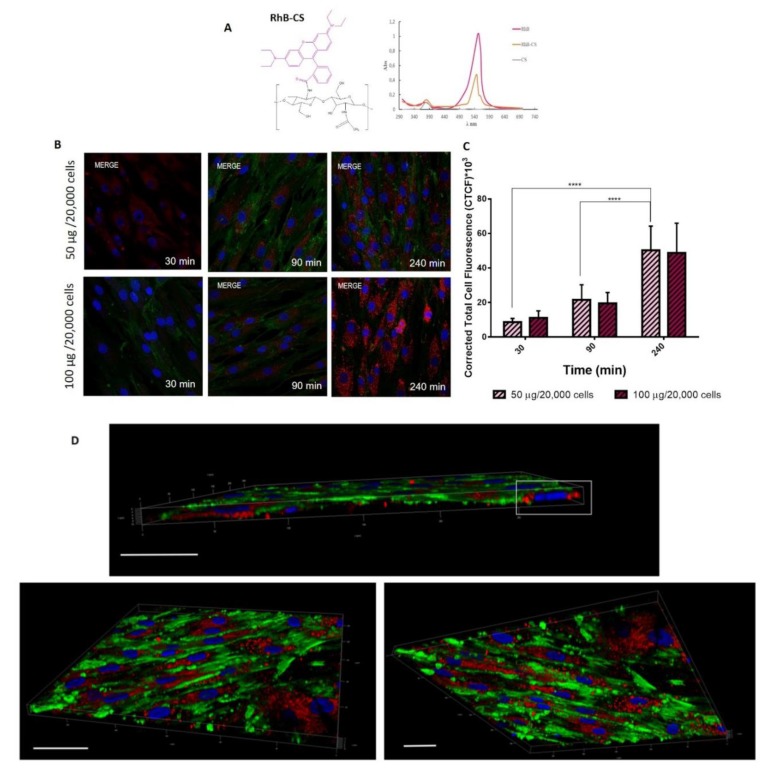
(**A**) RhB-conjugated CS chemical structure and its UV-vis spectrum compared to those of CS and RhB as such. (**B**) Confocal microscopy images of hMSCs after 30, 90 and 240 min of incubation at 37 °C with different concentration of CS-RhB/TPP NPs (50–100 μg/20,000 cells). Red fluorescent denotes RhB-CS/TPP NPs; nuclear blue fluorescence of DNA with Hoechst33258 dye; positive expression for CD44 on hMSCs membrane was detected by using anti-CD44 primary antibody and FITC labeled secondary antibody (green fluorescence), obj. mag 63X. (**C**) Red fluorescent intensity of CS-RhB/TPP NPs localized in the cell cytoplasm after 30, 90 and 240 min of incubation. Results are presented as mean ± SD (**** *p*-value < 0.0001). (**D**) 3D imaging of hMSC culture at 240 min (scale bar: 50 μm). hMSC was imaged in z-series on the Leica TCAAP8 confocal microscope and rendered in 3D with Imaris software. CS-RhB/TPP NPs localization in the cell cytosol is highlighted by the white square.

**Figure 7 ijms-20-06212-f007:**
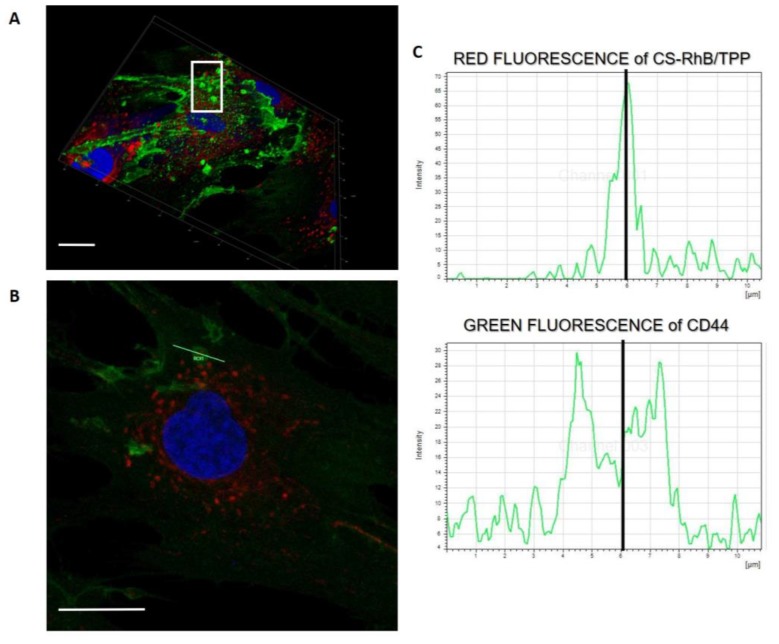
(**A**) 3D imaging of hMSC culture at 30 min. Cell membrane protrusions are highlighted by the white square (scale bar: 20 μm). (**B**) Detail of the endocytic vesicles formation: confocal image of hMSCs after 30 min incubation with CS-RhB/TPP NPs (100 µg/20,000 hMSCs), a yellow line, crossing a vesicle, was drawn (scale bar: 20 μm). (**C**) Analysis of the red and green fluorescence intensities across this line: graph I represents the red fluorescence intensity and graph II represents the green fluorescence intensity.

**Figure 8 ijms-20-06212-f008:**
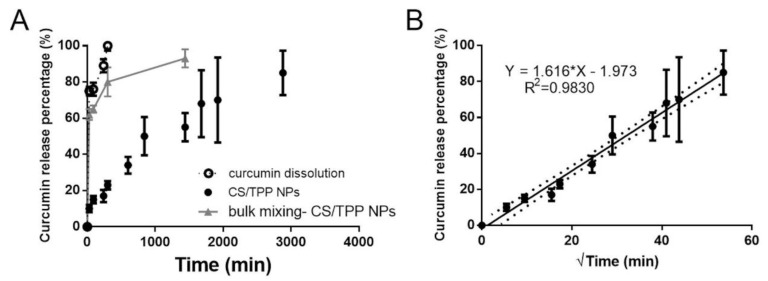
(**A**) In vitro CURC release profiles in PBS pH 7.4 (+CTAB, 0.5% *w*/*v*) at 37 °C from CS/TPP NPs prepared using the SHM-assisted method (•) and bulk mixing method (Δ). The CURC dissolution profile was reported as comparison (ο). (**B**) Higuchi’s model: equation and degree of fitting.

**Table 1 ijms-20-06212-t001:** Optimization of staggered herringbone micromixer (SHM)-assisted manufacturing process of CS/TPP NPs by the design of experiment (DoE): mean size, polydispersity index (PDI) and surface charge (ζ potential) of the different batches (Doe batch # (No.) 1–9); mean ± SD, *n* = 3 independent experiments..

Doe Batch #	DoE Level			Mean Size ± SD (nm)	PDI ± SD	ζ Potential ± SD (mV)
	[CS] (mg/mL)	[TPP] (mg/mL)	TFR (mL/min)			
1	2.5 (+1)	0.5 (+1)	12 (+1)	162 ± 28	0.53 ± 0.11	+18.9 ± 0.6
2	2.5 (+1)	0.5 (+1)	5 (−1)	372 ± 42	0.37 ± 0.02	+34.6 ± 1.0
3	2.5 (+1)	0.3 (−1)	12 (+1)	40 ± 12	0.65 ± 0.07	+28.9 ± 4.6
4	2 (−1)	0.5 (+1)	12 (+1)	69 ± 4	0.33 ± 0.01	+20.9 ± 2.1
5	2 (−1)	0.3 (−1)	12 (+1)	39 ± 12	0.51 ± 0.07	+24.5 ± 6.5
6	2 (−1)	0.5 (+1)	5 (−1)	305 ± 78	0.35 ± 0.07	+27.4 ± 0.9
7	2.5 (+1)	0.3 (−1)	5 (−1)	225 ± 79	0.46 ± 0.03	+30.5 ± 4.1
8	2 (−1)	0.3 (−1)	5 (−1)	245 ± 26	0.31 ± 0.04	+32.0 ± 2.9
9	2.25 (0)	0.4 (0)	8.5 (0)	119 ± 10	0.40 ± 0.02	+25.5 ± 3.0

**Table 2 ijms-20-06212-t002:** Summary of the physicochemical properties, CS recovery and curcumin (CURC) encapsulation efficiency of the CURC loaded CS/TPP NPs obtained by the SHM device and conventional bulk mixing method.

Preparation Method	NPs Mean Size ± SD (nm)	PDI ± SD	ζ Potential ± SD (mV)	CS Recovery	CURC Encapsulation Efficiency (%)
**SHM**	157 ± 41	0.31 ± 0.02	+25.0 ± 1.3	48.7 ± 3.6	74.81 ± 2.78
**Bulk mixing**	113 ± 15	0.36 ± 0.05	+17.1 ± 3.7	35.2 ± 6.6	37.12 ± 10.34

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
