# Peer review of "Staggered Herringbone Microfluid Device for the Manufacturing of Chitosan/TPP Nanoparticles: Systematic Optimization and Preliminary Biological Evaluation"

_ijms, 2019, doi:10.3390/ijms20246212_

Round 1

Reviewer 1 Report

This paper presents a interesting study of CS/TPP nanoparticles synthesized by ionic gelation method. The authors obtain the CS/TPP NPs using a microfluidic approach based on a staggered herringbone micromixer and they make a systematic evaluation of the factors affecting NPs physical properties. Moreover, the authors perform a preliminary biological evaluation and load successfully curcumin into CS/TPP NPs. The experimental results seem to be sound and the paper has enough quality to be published in International Journal of Molecular Sciences after some revision. - Throughout the text: the authors express the value of size, PDI, zeta potentia, Nagg, etc. with too significant figures. For example, it makes no sense to express the size of the nanoparticles with 5 significant figures, 161.73 nm, when the error made is +/-28.37 nm. In my opinion, the size would be 162+/-28 nm. The authors have to express the results with the adequate significant figures. - Lines 304-306: the authors say that the limiting step of the NPs recovery is the centrifugation indicating that a increase of centrifugation force or a change in the purification may improve the recovery. The authors should explain this fact in more detail. - Figure 4B: it would be more interesting to see the results of TFR 5mL / min instead of 12 mL/min. -Lines 323-328: the data of citotoxicity test should be reported. - Figure 4 B: a different absorbance of the UV-vis spectra of CS-RhB and RhB is observed. Is rhodamine concentration the same for both spectra? If so, the authors should explain this change in absorbance, and if not, they should indicate the rhodamine concentration in each. - Figure 4 C: there is a mistake in this figure. Do some photographs correspond to 50 ug NPs/20000 cells and others 100 ug/20000 cells? - Lines 421-423: in order to mimic the physiological condition the authors have used PBS at pH 7.4 with CTAB. It is surprising that the authors used such a toxic surfactant to simulate physiological conditions. They should have used a less toxic surfactant. Why have the authors select CTAB? By the way, CTAB does not appear in the materials section. - Lines 426-427: the authors have to explain the Higuchi’s model and discuss the figure 7B in detail. -Lines 430-431: The authors say that the release of CURC is completed at 24 h but this is only true for the bulk mixing-CS/TPP NPs? They should explain the difference observed in the CURC release profiles from CS/TPP NPs and bulk mixing-CS/TPP NPs. -Lines 505-512: Why do the authors use 10 mM NaCl aqueous solution in the measurements of the size and the surface charge of CS/TPP NPs? -Lines 528: there is a mistake in eq. 4, CNP appears instead of CNPs.. The authors should indicate that Rh represents in eq. 4. - References 7, 18, 22, 38, 40 and 46: the pages do not appear. - The paper corresponding to ref.15 is poorly referenced.”..., 2018, 48, 323-332...”

Author Response

Reviewer’ comments (in black) and Authors’ answers (in red).

This paper presents a interesting study of CS/TPP nanoparticles synthesized by ionic gelation method. The authors obtain the CS/TPP NPs using a microfluidic approach based on a staggered herringbone micromixer and they make a systematic evaluation of the factors affecting NPs physical properties. Moreover, the authors perform a preliminary biological evaluation and load successfully curcumin into CS/TPP NPs. The experimental results seem to be sound and the paper has enough quality to be published in International Journal of Molecular Sciences after some revision.

- Throughout the text: the authors express the value of size, PDI, zeta potentia, Nagg, etc. with too significant figures. For example, it makes no sense to express the size of the nanoparticles with 5 significant figures, 161.73 nm, when the error made is +/-28.37 nm. In my opinion, the size would be 162+/-28 nm. The authors have to express the results with the adequate significant figures.

Data have been expressed accordingly with the reviewer suggestion and a new table has been added in the reviewed manuscript. Data inside the text have been properly corrected.

- Lines 304-306: the authors say that the limiting step of the NPs recovery is the centrifugation indicating that a increase of centrifugation force or a change in the purification may improve the recovery. The authors should explain this fact in more detail.

Use of centrifugal force is a common technique in size-selective particle separation. The particle motion in the centrifugal field is described by i) the particle settling velocity, ii) the distance from the axis of rotation, iii) angular velocity, and iv) the sedimentation coefficient which depends on particle diameter, and particle and liquid densities, respectively, and v) the viscosity of liquid. Therefore NPs move toward the centrifugal field at different velocities depending on their size. When the NPs size are reduced an increase of the centrifugation force is required to recover them.

In the revised manuscript (Lines 305-308) the concept has been more clearly rephrased and the complete explanation was maintained in the Conclusions section.

- Figure 4B: it would be more interesting to see the results of TFR 5mL / min instead of 12 mL/min.

As requested by the reviewer data were implemented showing the results of 5 mL/min TFR (Figure 4C) and discussion has been implemented in the revised manuscript.

-Lines 323-328: the data of citotoxicity test should be reported.

In the revised manuscript cytotoxicity test results are included (Figure 5B) in the text as suggested by the reviewer.

- Figure 4 B: a different absorbance of the UV-vis spectra of CS-RhB and RhB is observed. Is rhodamine concentration the same for both spectra? If so, the authors should explain this change in absorbance, and if not, they should indicate the rhodamine concentration in each.

The concentrations are the same for all samples CS, RhB and RhB-CS (5 µg/ml), however the RhB concentration is different namely 5 µg/ml for RhB solution as such and 1.056 µg/mL for the fluorescent polymer depending on the RhB labelling degree on the CS.

In the revised manuscript the UV-vis procedure was better explained in the Methods section (Lines 609-611).

- Figure 4 C: there is a mistake in this figure. Do some photographs correspond to 50 ug NPs/20000 cells and others 100 ug/20000 cells?

We apologize for the mistake, Figure 4C (Figure 5C in the revised manuscript) has been corrected.

- Lines 421-423: in order to mimic the physiological condition the authors have used PBS at pH 7.4 with CTAB. It is surprising that the authors used such a toxic surfactant to simulate physiological conditions. They should have used a less toxic surfactant. Why have the authors select CTAB? By the way, CTAB does not appear in the materials section.

During the in vitro release test PBS at pH 7.4 was used in order to mimic physiological condition; CTAB was added to the buffer in order to ensure good solubility and stability of curcumin at that pH during the release test period [Benediktsdottir BE et al. 2015; Tønnesen HH, Karlsen J 1985]. To better explain the issue, as suggested by the reviewer, text has been rephrased and the reference has been added.

Furthermore, CTAB has been added in the Material Section.

- Lines 426-427: the authors have to explain the Higuchi’s model and discuss the figure 7B in detail. –

Authors agree with the reviewer, Higuchi model and Figure 8B (ex-7B) have been more deeply discussed in the revised manuscript (Lines 448-456).

Lines 430-431: The authors say that the release of CURC is completed at 24 h but this is only true for the bulk mixing-CS/TPP NPs? They should explain the difference observed in the CURC release profiles from CS/TPP NPs and bulk mixing-CS/TPP NPs.

The revised manuscript has been modified and implemented according to the reviewer’s suggestion (Line 441-448).

-Lines 505-512: Why do the authors use 10 mM NaCl aqueous solution in the measurements of the size and the surface charge of CS/TPP NPs?

Zeta potential is determined by two separate effects: charge at the shear plane and free ion concentration (e.g., dissolved salts). In order to monitor the change in zeta potential, one normally adds just enough dissolved salt to keep its effect constant. Samples should be prepared in a low ionic strength medium; 10 mM NaCl is highly recommended. Dilution in an electrolyte, such as NaCl, should ensure that any changes in the zeta potential values obtained were not due to conductivity differences (Phil. Trans. R. Soc. A (2010) 368, 4439–445; https://ncl.cancer.gov/sites/default/files/protocols/NCL_Method_PCC-2.pdf)

-Lines 528: there is a mistake in eq. 4, CNP appears instead of CNPs.. The authors should indicate that Rh represents in eq. 4.

We apologized for the mistake, eq 4 is corrected in the revised manuscript.

- References 7, 18, 22, 38, 40 and 46: the pages do not appear. - The paper corresponding to ref.15 is poorly referenced.”..., 2018, 48, 323-332...”

In the revised manuscript References have been properly added or corrected.

Reviewer 2 Report

Review on manuscript ijms-652739

The article is about the fabrication of well-known Chit-TPP NPs using microfluid device. 

The main problem is the novelty of the reserch work. Both the chit-Tpp as well as the curcumin encapsulation are widely pubished. The authors can not present any interesting and convincing results on chit-tpp system using flow system. 

I do not recommend this article for publication in this vey good journal with high impact. 

Author Response

Reviewer’ comments (in black) and Authors’ answers (in red).

The main problem is the novelty of the reserch work. Both the chit-Tpp as well as the curcumin encapsulation are widely pubished. The authors can not present any interesting and convincing results on chit-tpp system using flow system. 

One of the main hurdle to effective clinical translation of nanoparticles-based therapy is the lacking of satisfactory and efficient manufacturing techniques.

The novelty of the research paper is the systematic optimization of an innovative, reliable and reproducible microfluidics technique based on a staggered herringbone geometry for chitosan nanoparticles (NPs) synthesis, by exploiting polyelectrolytic interactions with the safe cross-linker tripolyphosphate (TPP). Used for the first time for chitosan NPs preparation, this microfluidic approach could address the lack of medium/large scale manufacturing of chitosan NPs.

Curcumin, poor water-soluble drug, was chosen as challenging model drug to be encapsulated into CS, a hydrophilic polymer. Feasibility of the new preparation method was also revealed by the comparison of curcumin encapsulation efficiency and release profiles of NPs made by the proposed microfluidic technique and the traditional bulk mixing method.

The set-up microfluidics-based technique revealed to be able to encapsulate curcumin with a much higher payload, a reduced burst effect and controlled release was obtained for the hydrophobic drug setting out a good dispersion of the drug inside the nanoparticles.

Design of experiment (DoE) was applied to identify the process parameter affecting the NPs physical properties; deep NPs physical characterization was performed assessing size, size distribution, surface charge, NPs compactness and number of chitosan moieties assembled in NPs. Preliminary in vitro studies concerning the NPs interaction with mesenchymal stem cells from human bone marrow were performed.

Reviewer 3 Report

The authors refer that a comparison between the conventional bulk mixing method and the microfluidics-assisted method was performed.

A summary Table showing the obtained results in terms of I.E. of curcumin, mean size, charge and chitosan recovery should be included.

Minor comments

Please check the references in the text in all document:

Example : “focusing (HFF)[3,27].” - a space is missing between the text and the references

Author Response

Reviewer’s comments (in black) and Authors’ answers (in red)

The authors refer that a comparison between the conventional bulk mixing method and the microfluidics-assisted method was performed.

A summary Table showing the obtained results in terms of I.E. of curcumin, mean size, charge and chitosan recovery should be included.

As suggested by the reviewer a summary Table has been added.

Minor comments

Please check the references in the text in all document:

Example : “focusing (HFF)[3,27].” - a space is missing between the text and the references

The text has been checked for typing errors.

Round 2

Reviewer 2 Report

After corrections I accept this article for publication in this version.